# Potential herd protection against *Plasmodium falciparum* infections conferred by mass antimalarial drug administrations

Daniel M Parker[1]*, Sai Thein Than Tun[2,3], Lisa J White[2,3], Ladda Kajeechiwa[4], May Myo Thwin[4], Jordi Landier[5], Victor Chaumeau[3,4,6,7], Vincent Corbel[7], Arjen M Dondorp[2,3], Lorenz von Seidlein[2,3], Nicholas J White[2,3], Richard J Maude[2,3,8], François Nosten[3,4]

[1]Department of Population Health and Disease Prevention, University of California, Irvine, United States; [2]Mahidol-Oxford Tropical Medicine Research Unit, Faculty of Tropical Medicine, Mahidol University, Nakhon Pathom, Thailand; [3]Centre for Tropical Medicine and Global Health, Nuffield Department of Medicine, University of Oxford, Oxford, United kingdom; [4]Shoklo Malaria Research Unit, Mahidol-Oxford Tropical Medicine Research Unit, Faculty of Tropical Medicine, Mahidol University, Nakhon Pathom, Thailand; [5]Institut de Recherche pour le Développement, University of Montpellier, Montpellier, France; [6]Centre Hospitalier Universitaire de Montpellier, Montpellier, France; [7]Maladies Infectieuses et Vecteurs, Ecologie, Génétique, Evolution et Contrôle IRD 224-CNRS 5290UM1-UM2, Institut de Recherche pour le Développement (IRD), University of Montpellier, Montpellier, France; [8]Harvard TH Chan School of Public Health, Harvard University, Harvard, United States

*For correspondence:
dparker1@uci.edu

**Abstract** The global malaria burden has decreased over the last decade and many nations are attempting elimination. Asymptomatic malaria infections are not normally diagnosed or treated, posing a major hurdle for elimination efforts. One solution to this problem is mass drug administration (MDA), with success depending on adequate population participation. Here, we present a detailed spatial and temporal analysis of malaria episodes and asymptomatic infections in four villages undergoing MDA in Myanmar. In this study, individuals from neighborhoods with low MDA adherence had 2.85 times the odds of having a malaria episode post-MDA in comparison to those from high adherence neighborhoods, regardless of individual participation, suggesting a herd effect. High mosquito biting rates, living in a house with someone else with malaria, or having an asymptomatic malaria infection were also predictors of clinical episodes. Spatial clustering of non-adherence to MDA, even in villages with high overall participation, may frustrate elimination efforts.
DOI: https://doi.org/10.7554/eLife.41023.001

## Introduction

Mass drug administration (MDA) is the provision of medications to entire target populations and the approach has been used for many infectious diseases, including lymphatic filariasis, soil-transmitted helminths, onchocerciasis, schistosomiasis, and trachoma (*Keenan et al., 2013*). MDA has historically been used for *P. falciparum* malaria (*Poirot et al., 2013*) and has recently been trialed in several

**eLife digest** The global burden of malaria has decreased over the last decade. Many countries now aim to banish malaria. One obstacle to elimination is people who carry malaria parasites without showing symptoms. These asymptomatic people are unlikely to be diagnosed and treated and may contribute to further spread of malaria. One way to clear all malaria infections would be to ask everyone in a community to take antimalarial drugs at the same time, even if they do not feel ill. This tactic is most likely to work in communities that are already reducing malaria infections by other means. For example, by treating symptomatic people and using bed nets to prevent bites from malaria-infected mosquitos.

Several studies have shown that mass drug administration is a promising approach to reduce malaria infections. But its success depends on enough people participating. If enough community members take antimalarial drugs, then even those who cannot participate, such as young children or pregnant women, should be less likely to get malaria. This is called the herd effect.

Now, Parker et al. demonstrate that mass antimalarial drug administration reduces infections with malaria caused by the parasite *Plasmodium falciparum*. The analysis looked at malaria infections among residents of four villages in the Kayin State of Myanmar that used mass antimalarial drug administration. People who lived in neighborhoods with high participation in mass drug administration were almost three times less likely to get malaria than people who lived in communities with low participation. Even people who did not take part benefited.

The analysis suggests that mass antimalaria drug administration benefits individuals and their communities if enough people take part. To be successful, malaria elimination programs that wish to use mass drug administration should approach communities in a way that encourages high levels of participation.

DOI: https://doi.org/10.7554/eLife.41023.002

locations in Africa (*Gitaka et al., 2017*; *Mwesigwa et al., 2018*; *Shekalaghe et al., 2011*) and Asia (*Manning et al., 2018*; *Tripura et al., 2018*; *Nguyen et al., 2018*; *Pongvongsa et al., 2018*; *Landier et al., 2017a*). It is being considered by several nations as a tool (to be used in unison with other interventions) for elimination (*World Health Organization, 2017*; *Zuber and Takala-Harrison, 2018*), and has already been implemented as an operational strategy in at least one modern setting (*Parker et al., 2017*; *Landier et al., 2018a*).

Given that drug pressure (through provision of antimalarial drugs) provides a survival advantage for resistant parasites, there has been some hesitance in using MDA for malaria. One historical malaria eradication campaign relied on the inclusion of sub-therapeutic levels of antimalarials distributed in table salt across large populations (*Pinotti et al., 1955*). This program likely led to the emergence of parasite resistance in the same regions (*Wootton et al., 2002*) and this has in part led to hesitance among some institutions (i.e. the World Health Organization and ministries of health) to implement MDA for malaria (*World Health Organization, 2015a*). MDA should ideally be used in settings with strong public health infrastructure, including easy access to diagnosis and treatment; an up-to-date and responsive surveillance system; and effective community engagement. Used appropriately, MDA can quickly reduce or eliminate parasite reservoirs and can act as a catalyst for subregional elimination of *P. falciparum* malaria (*Landier et al., 2018a*).

While antimalarials are usually administered following diagnosis (confirmed or presumed) or used as a prophylactic, MDA is used because of an intended population- or community-level effect. The rationale is that the transmission potential or reproductive rate of malaria is so high that a sufficient amount of the parasite reservoir needs to be removed in order to disrupt transmission. This group-level effect is also referred to as a 'herd effect' (*John and Samuel, 2000*; *Pollard et al., 2015*) and the concept applies to most communicable diseases. If a sufficient amount of the population participates in MDA, transmission chains cannot be sustained and transmission will cease, ultimately leading to a reduction in malaria morbidity and mortality (*World Health Organization, 2017*). There is likely to be a context-specific critical threshold for MDA coverage, below which the reduction of the parasite reservoir is not sufficient to halt ongoing transmission. Some literature has suggested that at least 80% coverage and adherence of MDA in the targeted population is necessary in order for

the MDA to be successful (*World Health Organization, 2015b*). If the aim for antimalarial MDA is to interrupt transmission, the notion of a herd effect providing additional levels of population protection is plausible but has not been examined empirically (*Cotter et al., 2013*).

Drawing from detailed micro-epidemiological and spatial data from an MDA trial in Kayin State, Myanmar (*Figure 1*), we describe geographic and epidemiological patterns of clinical and subclinical *P. falciparum* malaria in villages undergoing MDA. We investigate associations between individual- and group-level participation in MDA (potential direct and indirect effects, respectively); subclinical infections; and clinical episodes of *P. falciparum* post-MDA. Such empirical research is important for providing an evidence base for further research, intervention, and policy work.

## Results

3229 villagers (1689 male) were included in this study. During the study period, 80 study participants were diagnosed with clinical *P. falciparum*. 201 participants were found to have *P. falciparum* by uPCR. 325 uPCR-positive participants had Plasmodium infections not identifiable at the species level

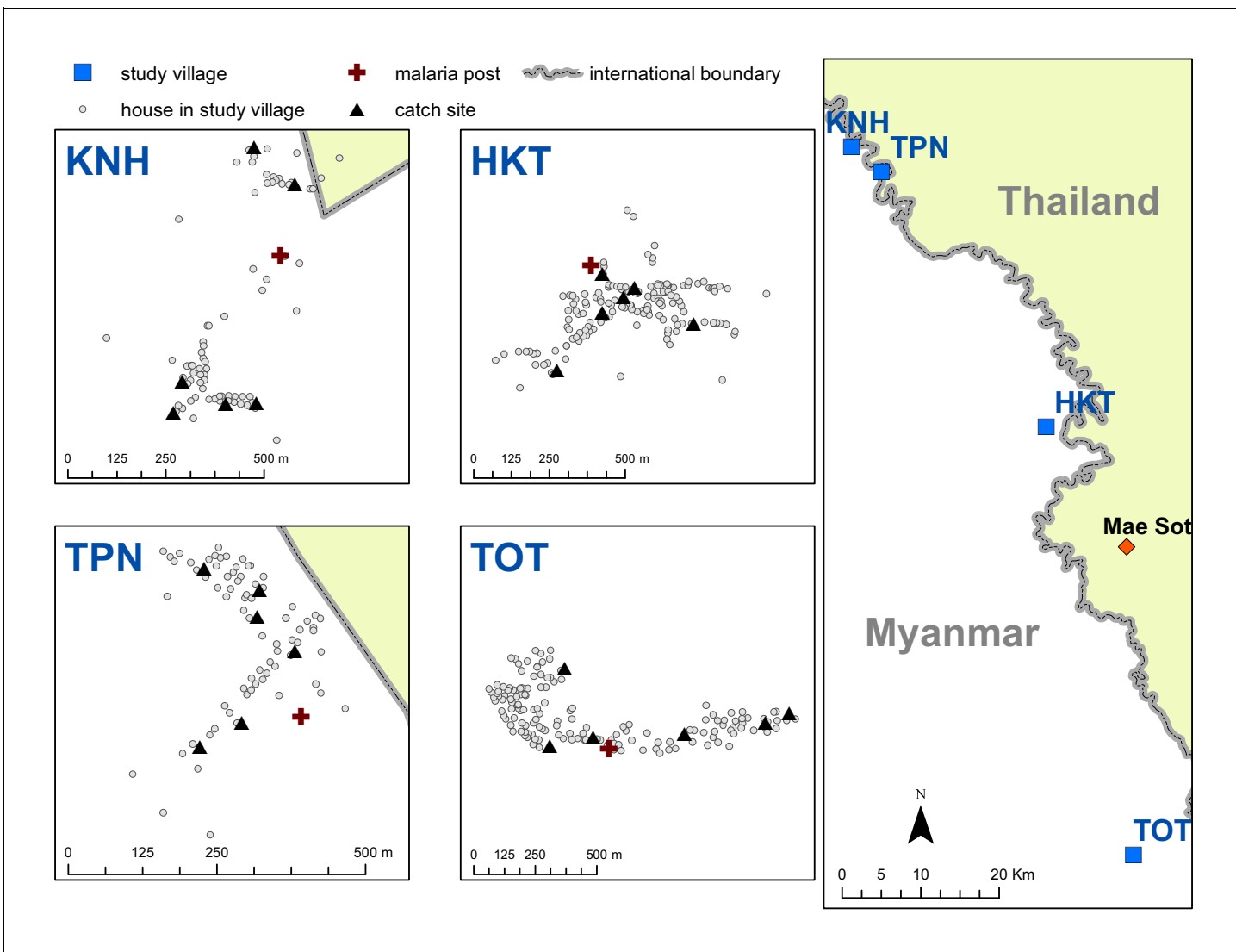

**Figure 1.** Map indicating the locations of the study villages along the Myanmar-Thailand border; and the distribution of houses, mosquito catch sites and malaria posts within study sites.

DOI: https://doi.org/10.7554/eLife.41023.003

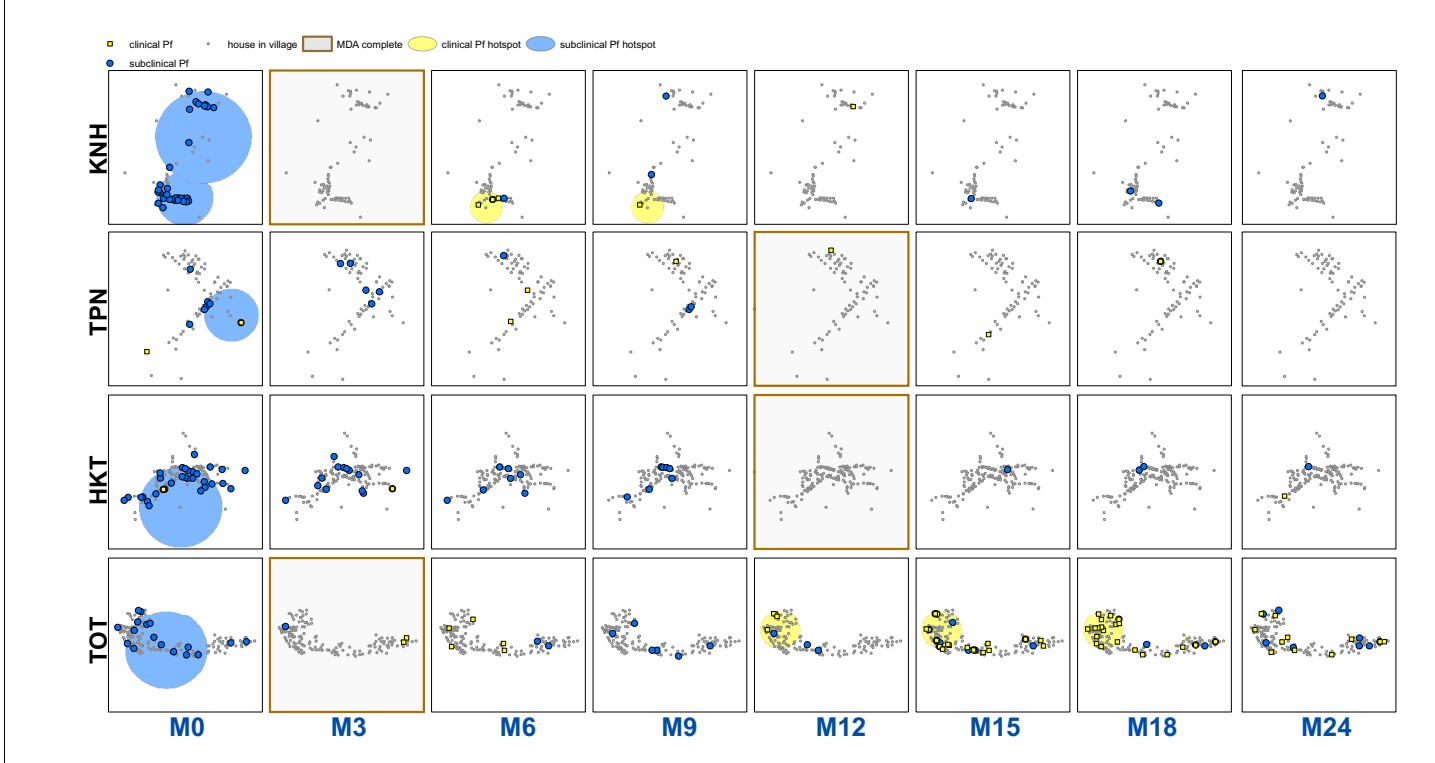

**Figure 2.** Clinical *P.falciparum* episodes (yellow square points) and uPCR-detected *P. falciparum* infections (blue dots) at house level over time (by survey month; month 0 (M0) through month 24 (M24) for each of the four study villages. Statistically significant clusters (detected using SaTScan) are indicated for both clinical episodes (underlying yellow circles) and uPCR-detected infections (underlying blue circles). Grey points indicate house locations for houses with no infections or episodes in a given time. In the maps clinical episodes are aggregated to align with surveys (i.e. M1, M2 and M3 aggregated into M3), though they were recorded and analyzed by individual month. The study was conducted from May 2013 through June 2015 (KNH began in June; TPN in May; HKT in July; and TOT in May of 2013).

DOI: https://doi.org/10.7554/eLife.41023.004

and thus were not included in these analyses. Total numbers of clinical episodes and uPCR-detected infections were higher than the total number of infected individuals because some participants had multiple infections.

After MDA, the vast majority of clinical *P. falciparum* episodes occurred in only one of the study villages (TOT). 66 out of the 80 participants who had a clinical *P. falciparum* episode were from TOT village (three from HKT, seven from TPN and four from KNH).

Eleven of the 80 participants (14%) who had a clinical *P. falciparum* episode during the study period had repeated clinical episodes and 19 of the 80 (24%) participants who had a clinical episode were found to have a uPCR-detected *P. falciparum* infection in at least one of the surveys. uPCR-detected *P. falciparum* infections were more prevalent in males than females (UOR: 2.03; CI: 1.50–2.76).

## Spatiotemporal patterns in clinical episodes, uPCR-detected infections, and MDA adherence

uPCR-detected *P. falciparum* infections were widespread in all villages at baseline (*Figure 2*). These infections were significantly reduced following MDA in all villages. The prevalence of uPCR-detected *P. falciparum* infections had reduced in two control villages (villages TPN and HKT) prior to MDA.

There were statistically significant clusters of uPCR-detected *P. falciparum* infections in each village at baseline but subsequently no significant clusters were detected (*Figure 2*). Clusters of clinical *P. falciparum* episodes occurred in two villages (KNH and TOT). The cluster in KNH occurred from M5 through M7 but included only four episodes. There were two separate clusters in village TOT. A cluster in the western portion of the village began in M12 and lasted until M18 (with a total of 35

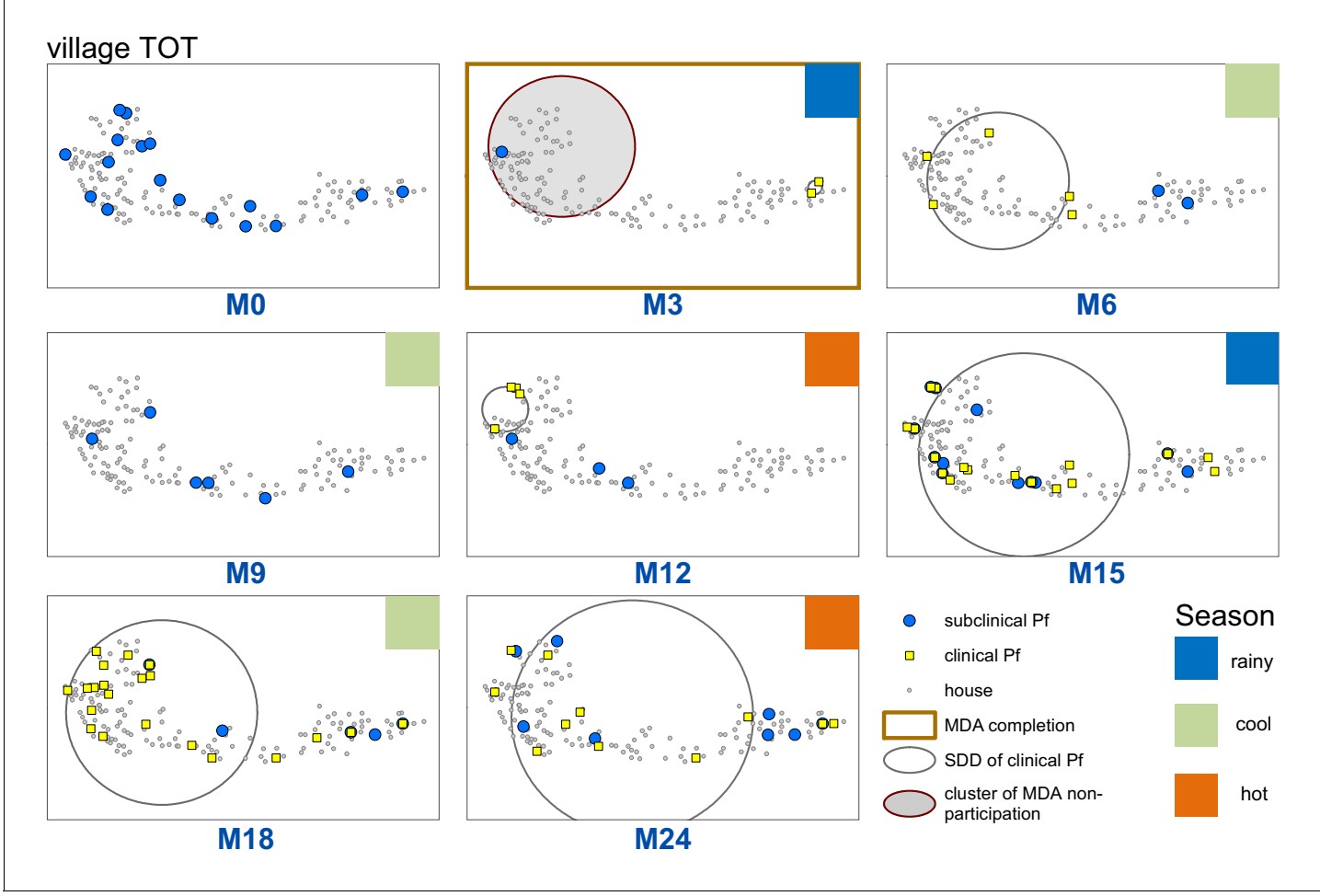

**Figure 3.** Spatiotemporal distribution of clinical *P.falciparum* episodes (yellow square points), uPCR-detected *P. falciparum* infections (blue dots), and a cluster of non-participation in MDA (grey circle/ochre border, detected using SatScan) in TOT village. Season is indicated by colored squares in the top right corner of each map. A measure of the spread of clinical *P. falciparum* cases is given by the standard distance deviation ('SDD'), indicated by the hollow circle with dark grey outline. One standard deviation is shown, indicating that roughly 68% of all cases lie inside of the circle. After MDA (M3), clinical episodes began occurring in the westernmost portion of the village. By month 15 (M15), clinical episodes were occurring throughout the village.
DOI: https://doi.org/10.7554/eLife.41023.005

episodes). A single-house cluster occurred in the eastern portion of the village (M15 through M18) with five episodes among four house members (2 in a 10 yo male, 1 in a 48 yo male, 1 in a 16 yo male, and 1 in a 48-year-old female).

There were significant clusters of non-participation in the MDAs in three of the study villages (TPN, HKT and TOT (*Appendix 1—figure 2*)). The non-participation cluster in TOT made up a large portion of the western part of the village and included 115 individuals not participating in the MDA (out of 919 total individuals in TOT). The non-participation clusters in HKT and TPN included 206 and 15 individuals respectively.

Sporadic clinical *P. falciparum* episodes occurred in village TOT following MDA (MDA was completed by M3), followed by a small outbreak beginning in M12 (*Figures 2* and *3*). The first clinical *P. falciparum* episodes during this outbreak occurred among villagers who lived in the cluster of non-MDA participation (*Figure 3*). By M15 the clinical episodes were occurring through much of the village (*Figure 3*).

Cumulative hazards plots of clinical *P. falciparum* episodes in village TOT illustrate the temporal patterns in infections according to neighborhood MDA non-adherence (aggregated into terciles) and individual participation in MDA (*Figure 4*). The proportion of individuals who had acquired a

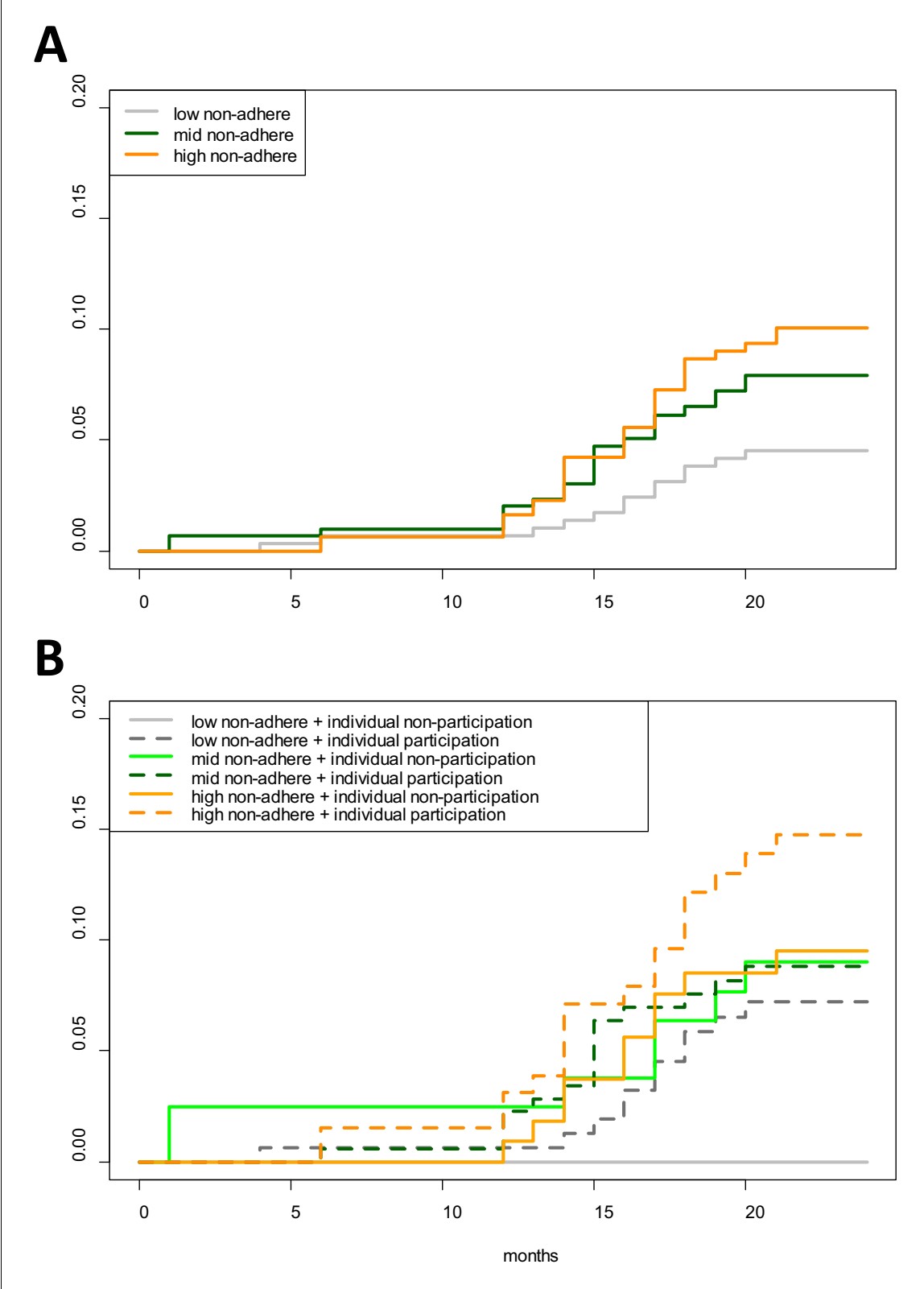

**Figure 4.** Cumulative hazard of having a clinical P. falciparum episode by MDA adherence. Cumulative hazard for clinical *P. falciparum* episodes in village TOT by (**A**) neighborhood MDA adherence ('low non-adhere' is a neighborhood with a low proportion of non-adherents; 'high non-adhere' is a neighborhood with a high proportion of non-adherents), (**B**) neighborhood adherence (same as in A) and individual adherence ('individual non-participation' indicates individuals who took no MDA while 'individual participation' indicates individuals who took at least 1 round of MDA). *Figure 4B*

*Figure 4 continued on next page*

*Figure 4 continued*

indicates that individuals who participated in MDA and lived in a neighborhood with low adherence had the highest risk of having a clinical episode post-MDA. Individuals who took no rounds of MDA but lived in a neighborhood with a high proportion of adherents had the lowest risk of acquiring a clinical episode post-MDA.

DOI: https://doi.org/10.7554/eLife.41023.006

clinical *P. falciparum* episode began consistently increasing in M12 for those living in either mid or high MDA non-adherence neighborhoods. *P. falciparum* episodes among low MDA non-adherence neighborhoods began increasing approximately 1 month after the increase in high non-adherence neighborhoods but never reached the level experienced in either mid or high MDA non-adherence neighborhoods. 4.4% of all individuals in low MDA non-adherence neighborhoods had at least one clinical *P. falciparum* episode by the end of the study period, in comparison to 7.6% in mid and 9.6% in high MDA non-adherence neighborhoods (log-rank test p-value=0.0485; *Figure 4A*).

The increase in clinical *P. falciparum* episodes in M12 also coincided with an increase in HBR in village TOT (*Appendix 1—figure 3*).

## Longitudinal multivariable analysis of clinical *P. falciparum* episodes

After MDA, clinical *P. falciparum* episodes in village TOT were most likely to occur among 5 to 14 year olds (AOR: 3.41; CI: 1.33–8.77, compared to 0 to 4 year olds) and participants who lived in a house with someone else who had a clinical *P. falciparum* episode during the same month (AOR: 3.43; CI: 1.52–7.72), after adjusting for other covariates (*Table 1*). Individuals who lived in a neighborhood with a high proportion of people who did not adhere to MDA had 2.8 times the odds of having a clinical episode (AOR: 2.85; CI: 1.28–6.37) compared to people who lived in neighborhoods where most people adhered to MDA (*Table 1*). The human biting rate was also associated with increased odds of having a clinical *P. falciparum* episode, with a 10% increase in odds for every one unit increase in HBR (AOR: 1.09; CI: 1.05–1.13).

**Table 1.** Multivariable mixed effects logistic regression for odds of having a clinical *P. falciparum* episode (village TOT only).
The model includes a random intercept for individual participants, with repeat observations occurring within individuals over the study period.

| Covariate | AOR | p-Value |
| --- | --- | --- |
| Age 0 to 4 | Comparison | |
| Age 5 to 14 | 3.41 (1.33–8.77) | 0.0104 |
| Age 15 plus | 2.17 (0.86–5.46) | 0.1053 |
| Female | Comparison | |
| Male | 1.19 (0.66–2.11) | 0.5612 |
| Participated in no rounds of MDA | Comparison | |
| Participated in MDA (at least one round) | 1.43 (0.73–2.78) | 0.2994 |
| No house member with clinical episode | comparison | |
| House member with clinical episode | 3.43 (1.52–7.72) | 0.0004 |
| Low neighborhood non-adherence to MDA | comparison | |
| Mid neighborhood non-adherence to MDA | 2.00 (0.87–4.60) | 0.0879 |
| High neighborhood non-adherence to MDA | 2.85 (1.28–6.37) | 0.0098 |
| Mean village HBR | 1.09 (1.05–1.13) | <0.0001 |
| Study month | 1.19 (1.10–1.27) | <0.0001 |

DOI: https://doi.org/10.7554/eLife.41023.008

**Table 2.** Table of predictor variables (covariates) used in regressions.

| Covariate | Level | Description |
|---|---|---|
| Age group | Individual | Ordinal; age groups: 0 to 4; 5 to 14; and 15 and above |
| Gender | Individual | Binary; male or female |
| Individual adherence to MDA | Individual | Binary; whether an individual participated in MDA or not (at least one full round) |
| Household member with clinical episode | Household | Binary; one if another house member had a clinical episode and 0 if not |
| Household member with uPCR-detected infection | Household | Binary; one if another house member had a uPCR-detected infection and 0 if not |
| Neighborhood MDA non-adherence | Household/ neighborhood | Ordinal (split into tertiles); proportion of people within 100 m radius who did not complete all three rounds of MDA |
| Human biting rate (HBR) | Village | Continuous; average number of bites per person per night |
| Study month | Village | Continuous; 1–26 (from May 2013 through June 2015); included as a control |

DOI: https://doi.org/10.7554/eLife.41023.007

## Discussion

The primary objective of this research was to look for a potential herd effect and at the impact of non-adherence with regard to MDA for *P. falciparum* malaria. There was an apparent group level effect from MDA adherence, suggesting herd protection and evident from three lines of evidence.

First, clinical episodes decreased among all groups for a longer period than the prophylactic effect (approximately 1 month) of the administered antimalarials. This group level protective effect from MDA was also evident in the rainy season following MDA (*Figure 3*) which corresponded to a surge in vector activity (*Appendix 1—figure 3*). Once the *P. falciparum* outbreak began (M12), there was a lag of approximately 1 month between the onset of clinical episodes in neighborhoods with mid and low MDA adherence and then occurred in neighborhoods with high MDA adherence (*Figure 4A*).

Second, neighborhoods with high MDA adherence never experienced the same levels of infection as those with mid or low MDA adherence (*Figure 4A*). Individuals who participated in MDA but lived in a neighborhood with low adherence had the highest risk of having a clinical episode whereas those who did not participate in MDA but lived in a neighborhood with high adherence had the lowest risk of having a clinical *P. falciparum* episode (*Figure 4B*). As has been described in other settings, this individual-level finding may be related to relative perceptions of risk; with potential complacency among individuals living in areas with lower levels of malaria (*Koenker et al., 2013*).

Third, the results from the multivariable logistic regression also suggest that living in a neighborhood with a high proportion of people who did not adhere to MDA was a significant risk factor for acquiring a clinical *P. falciparum* episode, after adjusting for individual MDA adherence and other important predictors of having a clinical episode (*Table 1*). To our knowledge, this is the first documentation of a herd effect conferred by MDA for *P. falciparum* malaria.

The increase in clinical *P. falciparum* episodes post-MDA also corresponded to an increase in village HBR. HBR also peaked in one other village (HKT) at the same time as in village TOT (*Appendix 1—figure 3*), but occurred in the absence of a detectable parasite reservoir and the HBR did not persist at high levels. Evidence also suggests that the MP in TOT was not functioning well in the first year of the study (reported in *Landier et al., 2017a*). The combination of a persisting parasite reservoir and persistently high HBR (from M13 – M18) in TOT likely explains the drastically different results between the study villages with regard to *P. falciparum* malaria elimination (*Figure 2*). A better functioning MP in TOT would likely have reduced the size of the outbreak.

Clustering of uPCR-detected *P. falciparum* infections across houses occurred for limited periods of time only prior to MDA (*Figure 2*). This clustering suggests that interventions such as reactive case detection would have resulted in the detection of extra cases (of both clinical and uPCR-detected *P. falciparum*) when searching within houses and occasionally in neighboring houses, but these would have only been a small proportion of all infections within the villages (*Parker et al., 2016*) and would not have halted transmission. Conversely, community based early diagnosis and

treatment and MDA with high participation, targeted at the village scale or larger, appear effective at reducing prevalence, incidence, and transmission of *P. falciparum* (*Landier et al., 2018b*).

There are several limitations to this work. Individuals who did not participate in MDA also did not participate in blood screenings immediately after MDA (i.e. M3 in village TOT). uPCR-detected infections are therefore likely to be underdiagnosed for these individuals, and it is likely that such infections clustered and overlapped with the clusters of non-adherence to MDA. While no genetic analyses were done with samples from the village, it is likely that these underdiagnosed infections, combined with the high HBR, led to a resurgence of clinical episodes. There is also evidence of a poorly functioning MP in this village, which could have led to undiagnosed clinical episodes, especially during the beginning of the study. Some infections are likely to be acquired outside of the village, leading to complex spatial patterns in infections that are mapped at the house level. Within-household clustering can be the result of within-household transmission, or shared exposure outside of the household or village among household members. Finally, these data come from a limited number of villages (total of 4), with analysis of *P. falciparum* episodes coming from the sole village that continued to have *P. falciparum* after MDA. Given that clinical *P. falciparum* episodes post-MDA were only possible to analyze in a single village, and that neighborhoods were not discrete and overlapped (100 m buffer around each house), a neighborhood-level effect was not included in this analysis. It is possible that the confidence intervals around the neighborhood MDA adherence variable are therefore too small and would not have been statistically significant had a neighborhood effect been included. This work would benefit from analyses with larger datasets.

This work has relevance with regard to further research and practice concerning MDA. While participation is obviously crucial to success, there is unlikely to be a single adherence proportion that can be applied in all situations. In this study, the elimination efforts were successful in three out of four villages even though one of those villages (HKT) had a similar overall adherence to the village with *P. falciparum* remaining after MDA (TOT). Likewise, this work points toward the need for considering spatial scale in MDA and in MDA adherence. Most current MDA trials and programs in the GMS and elsewhere consider a single village or community as the target unit. In some cases, especially when high prevalence villages are spatially clustered across a landscape, it may be necessary to target units above the village level (i.e. groups of villages).

## Materials and methods

### Study location and design

The study site consisted of four villages (KNH, TPN, HKT, and TOT) along the Myanmar-Thailand border, in Kayin (Karen) State, Myanmar (*Landier et al., 2017b*). The villages were selected based on *P. falciparum* malaria prevalence surveys using ultrasensitive quantitative PCR (uPCR) (*Imwong et al., 2015*) and were part of a MDA pilot study (*Landier et al., 2017b*). The northernmost village is approximately 105 km from the southernmost and the two closest villages, KNH and TPN are within 10 km of each other (*Figure 1*). The study was conducted from May 2013 through June 2015.

A full population census was completed in each of the four study villages at baseline May – June 2013. Everyone enumerated in the census was given a unique identification code. Geographic coordinates were collected for all houses in the four study villages and a unique identification code was assigned to each house. All individuals were then linked to their respective houses.

Blood surveys were conducted at baseline in each village, aiming to screen all individuals above an age of 9 months. Venous blood (3 mL) was drawn from each participant, transported to a central laboratory and analyzed using a highly sensitive quantitative PCR (uPCR) assay with a limit of detection of 22 parasites per mL. (*Imwong et al., 2014*). Infections detected through these blood screenings are hereafter referred to as uPCR-detected infections. Most (86%) uPCR-detected infections were subclinical (*Landier et al., 2017a*) and clinical infections were provided the standard treatment (see below).

A community-based malaria clinic (referred to as a malaria post or MP) was established in each village at the beginning of the project, as part of the malaria intervention. Village health workers were trained to diagnose malaria using rapid diagnostic tests (RDTs) and to treat RDT positive infections with dose based on weight and age. The ID code of each participant who self-presented at the MP

was recorded, along with RDT results, and these cases are hereafter referred to as clinical malaria episodes of either *P. falciparum* or *P. vivax*. Malaria episodes were treated with dihydroartemisinin-piperaquine (DHA + P) for *P. falciparum* and chloroquine for *P. vivax*. Radical cure for *P. vivax* was not provided because the absence of G6PD (Glucose-6-phosphate dehydrogenase) tests required to prevent hemolysis in G6PD individuals (*Bancone et al., 2014*; *Chu et al., 2017*).

MDAs were conducted in two villages at the beginning of the study (month 0, M0) and extended to the two control villages beginning in month 9 (M9). Restricted randomization was used to decide which villages received early or deferred MDA. MDA consisted of 3 days of DHA + P, with a single low dose of primaquine on the third day, repeated over three months (M0, M1, M2 for the first group and M9, M10, M11 for the control group). Follow-up blood surveys were conducted in each village every third month after M0 until M18. A final full blood survey was completed in each village at M24.

Mosquitoes were collected monthly using human landing catches to estimate the human biting rate (HBR). Mosquito catching teams were based at five sites (both indoors and outdoors) within each of the four study villages (total of 20 catch sites) for five consecutive nights during the study period M0 through M20. Mosquitoes were caught using glass tubes and later identified morphologically (*Ya-Umphan et al., 2017*).

The locations of study villages, MPs, catch sites, and village houses are indicated in *Figure 1*.

## Analysis

### Variables

All individuals recorded in the census with a house address in the four study villages were included in this analysis.

The data were aggregated into 1 month time steps and individuals were coded with a '1' for any month in which they presented at the village MP and were diagnosed with *P. falciparum*. Likewise, individuals who did not have a clinical episode within a given month were coded with a '0' for that respective month. Individuals who were ever diagnosed with a clinical episode or uPCR-detected *P. falciparum* infections were likewise coded as a '1' for analyses of having ever been detected by uPCR for an infection or having ever had a clinical episode during the study period.

Predictor variables (covariates) are listed in *Table 2*. Individual-level predictors included age group, gender, infection status, and adherence to MDA. Household-level predictor variables included a binary variable for whether or not another household member had a clinical episode and whether or not another household member had a uPCR-detected infection.

Neighborhood MDA non-adherence was calculated as the proportion of people who took no rounds of MDA within 100 m radius of each house in the study population. This proportion was calculated for each house in the study villages and non-adherence proportions were then attributed to individuals based on the house to which they were attributed.

The human biting rate (HBR) for primary vectors (*Anopheles minimus s.l.*, *An. maculatus s.l.*, and *An. dirus s.l.*) was calculated for each month.

### Exploratory spatial and temporal analyses

All predictor variables were explored in bivariate analyses. Unadjusted odds ratios (UOR) were calculated for binary predictors and Wilcox rank sum tests were calculated for continuous variables. Cumulative hazards curves were used to analyze temporal patterns in clinical episodes. uPCR-detected infections (from surveys) and clinical episodes (from the MPs) were mapped at the house level across time. Maps were created for each village and each survey time point (months 0 through 24: M0 – M24), with clinical episodes aggregated to align with surveys (i.e. M1, M2 and M3 aggregated into M3).

The weighted standard distance deviation (SDD) was used to visually analyze the distribution of clinical *P. falciparum* episodes for each survey time point in the one village with sufficient *P. falciparum* episodes (TOT) post-MDA. Clinical episodes were aggregated to align with survey time points (i.e. months 1 through 3 were aggregated and plotted in month 3). One SDD was calculated, corresponding to approximately 68% of all points falling inside of the resulting circle.

Scan statistics were used to test for clustering of uPCR-detected *P. falciparum* infections across survey months; clinical *P. falciparum* episodes across all months of the study period; and MDA non-

participation (*Kulldorff, 1997*). The scan statistics used a moving window (a circle) that centered on each point in the village, testing for the relative risk of cases given a population size within the circle in comparison to the risk outside of the circle (*Kulldorff, 1997*). The circle increased in size until it included half of the population and then moved to the next geographic reference point. For Plasmodium infections and malaria episodes the space-time discrete Poisson model was used, whereas for MDA participation a purely spatial Poisson model was used (*Kulldorff, 1997*) (as MDAs were completed within a 3-month time period).

## Multivariable logistic regression

A multivariable mixed effects logistic regression was used to estimate model adjusted odds ratios (AOR) and confidence intervals for predictor covariates with regard to the individual odds of having a clinical *P. falciparum* malaria episode. Since clinical *P. falciparum* episodes were almost exclusively limited to a single village post-MDA (village TOT), the analysis of clinical *P. falciparum* episodes focuses on this village alone.

Covariates for these models are listed in *Table 2* and included individual, household, neighborhood, and village-level predictors. Model and variable selection are described in detail in Appendix 1. Results for uPCR-detected *P. falciparum* infections are listed in Appendix 1. While this study focuses on *P. falciparum*, *P. vivax* data were also collected and are included in secondary analyses in the Appendix.

## Software

Exploratory statistics and regressions were calculated using R (version 3.4.3; https://cran.r-project.org/) and the 'epiR', 'lme4', and 'survival' packages. All maps were created using ArcGIS 10.5 (https://www.arcgis.com/). Exploratory spatial data analysis was conducted using ArcGIS 10.5 and SatScan v9.5 (https://www.satscan.org/). The neighborhood participation variable was created using ArcGIS and the Python programming language (version 3.5.2; https://www.python.org/).

## Ethics approval

The study protocol was reviewed and approved by the Oxford Tropical Research Ethics Committee (reference no. 1017–13 and 1015–13), the Tak Province Community Ethics Advisory Board (T-CAB), and by village committees in each of the four study villages. Survey and mass drug administration participation were voluntary and all participants provided written informed consent. Participants < 18 yo provided assent to participate, along with consent from their guardian(s). Potential participants received study information in their mother language (Karen or Burmese) both through community engagement activities on through one-on-one meetings. The ethics committees require consent to publish when individuals are identifiable. Since no individuals are identifiable through this publication, no consent to publish was obtained.

# Acknowledgements

We would like to thank the study communities in Kayin State, Myanmar for their participation, support and acceptance. We would also like to acknowledge the many staff members at Shoklo Malaria Research Unit and the Mahidol-Oxford Tropical Medicine Research Unit who made this project possible. This study is part of the larger 'Targeted Chemo-elimination (TCE) of Malaria (TME)' project, which is registered at ClinicalTrials.gov: NCT01872702 (https://clinicaltrials.gov/ct2/show/NCT01872702). Funding for the TME project was obtained from Wellcome Trust (101148/Z/13/Z) to Prof. Nicholas J. White and the Bill and Melinda Gates Foundation (OPP1081420) to Prof. Arjen M Dondorp. Sai Thein Than Tun is supported by the Wellcome Trust (grant no. 205240/Z/16/Z).

# Additional information

## Funding

| Funder | Grant reference number | Author |
| --- | --- | --- |
| Wellcome | 101148/Z/13/Z | Nicholas J White |

| Bill and Melinda Gates Foundation | OPP1081420 | Arjen M Dondorp |
|---|---|---|

The funders had no role in study design, data collection and interpretation, or the decision to submit the work for publication.

## Author contributions

Daniel M Parker, Conceptualization, Data curation, Formal analysis, Investigation, Visualization, Methodology, Writing—original draft, Project administration, Writing—review and editing; Sai Thein Than Tun, Formal analysis, Visualization, Methodology, Writing—original draft, Writing—review and editing; Lisa J White, Conceptualization, Visualization, Methodology, Writing—review and editing; Ladda Kajeechiwa, Data curation, Supervision, Project administration, Writing—review and editing; May Myo Thwin, Supervision, Project administration, Writing—review and editing; Jordi Landier, Data curation, Project administration, Writing—review and editing; Victor Chaumeau, Data curation, Investigation, Writing—review and editing; Vincent Corbel, Conceptualization, Data curation, Supervision, Investigation, Writing—review and editing; Arjen M Dondorp, Data curation, Supervision, Funding acquisition, Writing—review and editing; Lorenz von Seidlein, Conceptualization, Data curation, Supervision, Methodology, Writing—original draft, Writing—review and editing; Nicholas J White, Supervision, Investigation, Writing—review and editing; Richard J Maude, Conceptualization, Formal analysis, Visualization, Writing—review and editing; François Nosten, Supervision, Visualization, Project administration, Writing—review and editing

## Author ORCIDs

Daniel M Parker http://orcid.org/0000-0002-5352-7338
Jordi Landier https://orcid.org/0000-0001-8619-9775
Victor Chaumeau https://orcid.org/0000-0003-0171-2176
Arjen M Dondorp https://orcid.org/0000-0001-5190-2395
Nicholas J White https://orcid.org/0000-0002-1897-1978
François Nosten https://orcid.org/0000-0002-7951-0745

## Ethics

Clinical trial registration NCT01872702
Human subjects: This project was approved by the Oxford Tropical Research Ethics Committee (OxTREC: 1015-13; April 29, 2013) and the Tak Province Community Ethics Advisory Board (T-CAB). All participants provided written informed consent. Participants < 18 yo provided assent to participate, along with consent from their guardian(s).

## Decision letter and Author response

Decision letter https://doi.org/10.7554/eLife.41023.022
Author response https://doi.org/10.7554/eLife.41023.023

# Additional files

## Supplementary files

• Transparent reporting form
DOI: https://doi.org/10.7554/eLife.41023.009

## Data availability

The data used in these analyses are human subjects data from a sensitive population and organizational policy restricts data sharing for ethical and security considerations. Data can be accessed through the Data Access Committee at Mahidol Oxford Tropical Medicine Research Unit (MORU). The data sharing policy (including information on how to access the data) can be found here: http://www.tropmedres.ac/data-sharing.

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

## Appendix 1

DOI: https://doi.org/10.7554/eLife.41023.010

# Model selection and variable specification

## Regressions for clinical episodes

A multivariable mixed effects logistic regression was used to estimate model adjusted odds ratios (AOR) and confidence intervals for predictor covariates with regard to the individual odds of having a clinical *P. falciparum* malaria (*P. vivax* results provided below). Covariates for this model are listed in *Table 2* and included individual, household, neighbourhood, and village-level predictors.

Clinical *P. falciparum* episodes were almost exclusively limited to a single village post-MDA (village TOT), and the regression analyses of clinical *P. falciparum* episodes and uPCR-detected infections focuses on this village alone (whereas analyses for *P. vivax* included all four villages). The models for clinical episodes included a random intercept to account for repeated measures within individuals across the study period. During model selection, a random intercept was also tested for household but resulted in no significant model improvement and was therefore excluded in subsequent models.

Different specifications (continuous, binary, ordinal) for individual and neighborhood level MDA adherence were tested in initial models. The continuous specification for individual level MDA adherence was calculated as the proportion of all doses of dihydroartemisinin-piperaquine taken (total of 9 possible). A binary specification for whether or not an individual had taken one or more rounds of MDA was also tested. The continuous specification for neighborhood level MDA non-adherence was the proportion of all individuals within a 100 m radius who took no rounds of MDA. An ordinal specification was also tested, based off of the distribution of the data and categorized by tertiles (*Appendix 1—table 1*). The use of categorical or binary variables allowed us to test for temporal effects by different levels of neighborhood participation and by whether or not an individual participated in MDA via hazards curves for each category (*Figure 4*) These categorical specifications were chosen for the final models in the manuscript based on model fit statistics (AIC and BIC did not indicate that models with continuous predictors provided a better fit) and in order to maintain consistency with the visualizations in *Figure 4*. The model with continuous specifications for neighborhood MDA non-adherence and individual adherence to MDA is provided here for clinical *P. falciparum* episodes (*Appendix 1—table 2*).

**Appendix 1—table 1.** Terciles (lower, middle and upper 1/3) of MDA non-adherence (% taking no rounds of MDA) in TOT

| | |
|---|---|
| High non-adherence | >0.294 |
| Mid non-adherence | >0.20 and <0.294 |
| Low non-adherence | <0.20 |

DOI: https://doi.org/10.7554/eLife.41023.011

**Appendix 1—table 2.** Multivariable mixed effects logistic regression for odds of having a clinical *P. falciparum* episode. The model includes a random intercept for individual participants, with repeat observations occurring within individuals over the study period. Unlike the model in the main text (*Table 1*) neighborhood non-adherence to MDA and individual adherence to MDA are continuous covariates. Study month was included as a control (a linear specification was used, but polynomial specifications were also tested). The covariates for human biting rate (HBR) and having a house member with a clinical episode in the same month were specified as time-varying covariates.

| Covariate | AOR | p-value |
|---|---|---|

*Appendix 1—table 2 continued on next page*

*Appendix 1—table 2 continued*

| Covariate | AOR | p-value |
| --- | --- | --- |
| Age 0 to 4 | comparison | |
| Age 5 to 14 | 3.5 (1.3–9.0) | 0.0112 |
| Age 15 plus | 2.2 (0.9–5.6) | 0.1014 |
| female | comparison | |
| Male | 1.2 (0.7–2.2) | 0.5147 |
| no house member with clinical episode | comparison | |
| house member with clinical episode | 3.8 (1.8–7.7) | 0.0003 |
| proportion of MDA doses complete | 1.0 (1.0–1.1) | 0.4488 |
| proportion of non-adherers in neighborhood | 1.4 (1.0–1.8) | 0.0380 |
| mean village HBR | 1.1 (1.1–1.1) | <0.0001 |
| study month | 1.2 (1.1–1.3) | <0.0001 |

DOI: https://doi.org/10.7554/eLife.41023.012

The sensitivity of the model results with regard to the neighborhood MDA non-adherence buffer size was also tested. Neighborhood MDA-non-adherence was tested with varying buffer sizes, beginning with a radius of 20 m and going up to a radius of 220 m, by 40 m intervals. Neighborhood buffers from 60 m to 140 m were statistically significant and had comparable model adjusted odds ratios (at 60 m: AOR = 2.33; CI: 1.03–5.25; at 100 m: AOR = 2.84; CI: 1.29–6.27; at 140 m: AOR = 2.35; CI: 1.06–5.19). Small buffers (i.e. 20 m) included only one or very few households. At larger buffer sizes (≥180 m) most houses were included within a buffer (furthest distance between any two houses was approximately 1 km) leading to little spatial heterogeneity in participation across the village.

## Regressions for uPCR-detected infections

Multivariable logistic regressions were used to test for associations between predictor variables and the individual odds of being diagnosed by uPCR with a *P. falciparum* or *P. vivax* infection after MDA. Some individuals were detected with infections during multiple repeated screenings. However we are unable to differentiate between new and old infections with these data. We therefore tested for associations between potential risk factors and the log odds of having ever been detected by uPCR with an infection after MDA. Individuals who were ever diagnosed as infected with *P. falciparum* by uPCR were coded with a '1' and individuals who were never diagnosed with an infection were coded as a '0'. Since these models were not longitudinal and only have a single observation per individual, time-varying covariates (such as HBR) were not included. Predictor covariates in these models are listed and described in *Table 2*. Results for uPCR-detected *P. falciparum* are presented in *Appendix 1—table 3* and results for uPCR-detected *P. vivax* are presented in *Appendix 1—table 5*.

**Appendix 1—table 3.** Logistic regression for the odds of having a uPCR-detected *P. falciparum* infection after MDA. Individuals in the data were coded as having an infection if they were ever determined by uPCR to have an infection through blood screenings in full village blood surveys after MDA. Almost all *P. falciparum* episodes occurred in a single village (TOT) and the analysis for *P. falciparum* was only conducted on data from that village. HBR is not included in this regression as it varies across time.

| Covariate | AOR | p-value |
| --- | --- | --- |
| 0 to 4 | comparison | |
| five to 14 | 6.1 (1.1–113.9) | 0.0877 |

*Appendix 1—table 3 continued on next page*

*Appendix 1—table 3 continued*

| Covariate | AOR | p-value |
|---|---|---|
| 15 plus | 5.5 (1.1–99.3) | 0.1017 |
| female | comparison | |
| male | 1.3 (0.6–2.7) | 0.5217 |
| participated in no rounds of MDA | comparison | |
| participated in MDA (at least one round) | 0.4 (0.2–1.1) | 0.0955 |
| no house member with uPCR infection | comparison | |
| house member with uPCR infection | 1.1 (0.4–2.7) | 0.8039 |
| no house member with clinical episode | comparison | |
| house member with clinical episode | 1.7 (0.7–3.8) | 0.2029 |
| low neighborhood non-adherence to MDA | comparison | |
| mid neighborhood non-adherence to MDA | 1.1 (0.4–3.2) | 0.8743 |
| high neighborhood non-adherence to MDA | 2.6 (1.0–7.1) | 0.0488 |
| number of surveys attended | 1.3 (1.1–1.6) | 0.0075 |

DOI: https://doi.org/10.7554/eLife.41023.013

## Weighted *Standard Distance Deviation (SDD)*

The SDD gives an indication of how points deviate from the mean center. Weighted SDD are calculated by finding the geometric mean center of all points in a given time frame, weighted by the number of clinical episodes (*Chu et al., 2017*; *Ya-Umphan et al., 2017*). The SDD is frequently represented by a circular map layer centered on the weighted mean with the standard distance as the radius. The formula for the weighted SDD is:

$$SDD = \sqrt{\frac{\sum\limits_{i=1}^{n} w_i(x_i - X_{MC})^2}{\sum\limits_{i=1}^{n} w_i} + \frac{\sum\limits_{i=1}^{n} w_i(y_i - Y_{MC})^2}{\sum\limits_{i=1}^{n} w_i}}$$

where $w_i$ is the weight at point $i$;

$x_i$ and $y_i$ are geographic references for point $i$;

$\{X_{MC}, Y_{MC}\}$ is the geometric mean center (MC) for the features.

## Results from the analysis of clinical Plasmodium vivax episodes and uPCR-detected *P. vivax* infections

Of 3229 villagers included in the study, 216 participants were diagnosed with clinical *P. vivax*. 611 were found to have *P. vivax* infections by uPCR. *P. vivax* infections were more prevalent in males than in females (UOR: 1.7; CI: 1.4–2.0). *P. vivax* infections were reduced significantly following MDA in all villages but returned in subsequent months in most villages (as expected given that radical cure was not provided), with the exception of village TPN.

*P. vivax* infections were widespread throughout the study villages at baseline but no spatial clustering pattern was detected at month 0 (M0) (*Appendix 1—figure 1*). Village HKT had a cluster at M3; village KNH had one persistent cluster from M9 through M18; and village TOT had two clusters at M24 (*Appendix 1—figure 1*). Clusters of clinical *P. vivax* episodes also lingered in village KNH (M3 through M14 (17 episodes) and then M12 through M24 (six episodes)) and in village TOT (M12 through M21 (31 episodes) and M19 through M23 (18 episodes)). There was a cluster of clinical *P. vivax* episodes in village HKT during M23 and M24 (including 23 episodes).

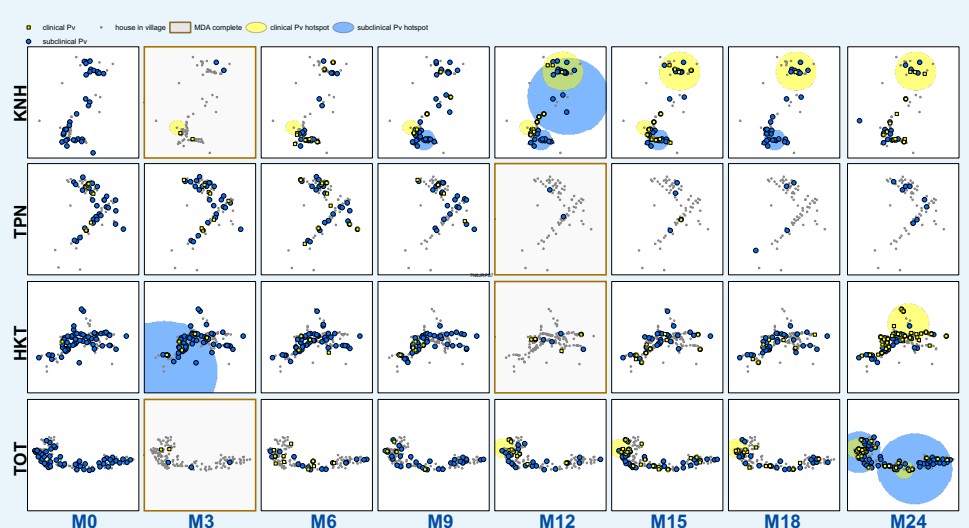

**Appendix 1—figure 1.** Clinical *P.vivax* episodes (yellow square points) and uPCR-detected *P. vivax* infections (blue dots) at house level over time for each of the four study villages. Statistically significant clusters (detected using SaTScan) are indicated for both clinical episodes (underlying yellow circles) and PCR-detected infections (underlying blue circles). MDA completion is indicated by the dark brown box bordering the survey time period (months 0 through 24, indicated as M0 – M24 along the x-axis). Blood stage *P. vivax* parasites were largely cleared following MDA. Since no accurate field diagnostic for glucose-6-phosphate dehydrogenase (G6PD) deficiency existed at the time, it was deemed unsafe to provide 7 or 14 days of the hypnozoitocidal drug primaquine. Many of the subsequent infections and clinical episodes are likely a result of re-emergence of hypnozoite-stage parasites rather than new *P. vivax* infections.

DOI: https://doi.org/10.7554/eLife.41023.014

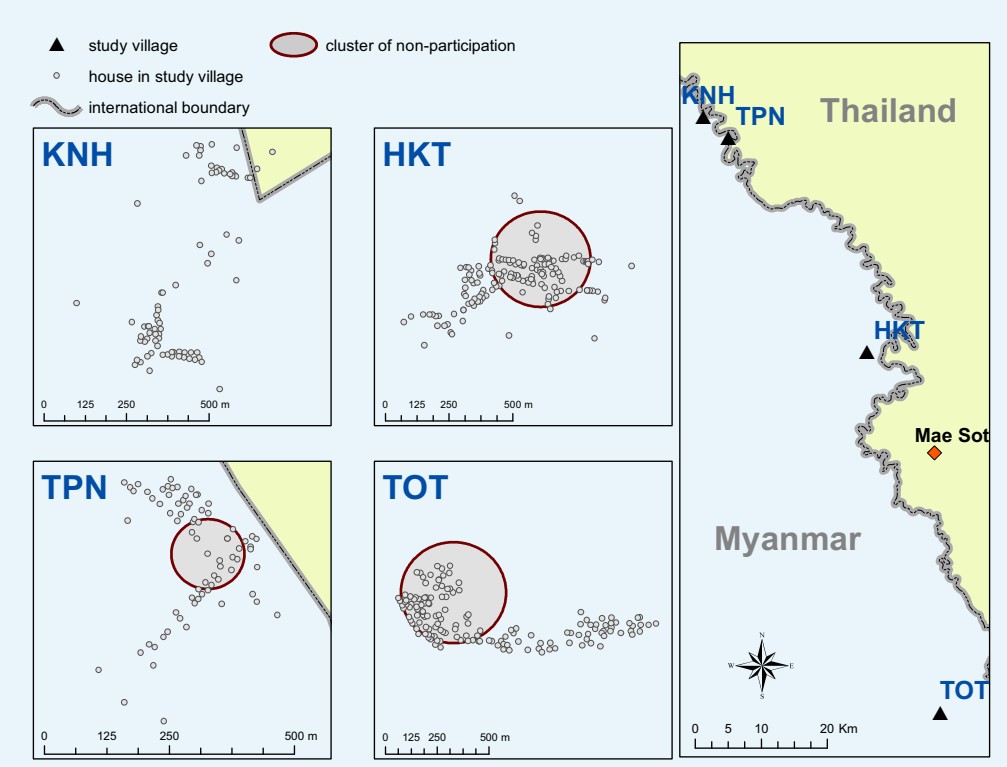

**Appendix 1— figure 2.** Spatial clusters (detecting using SatScan) of non-participation in MDA.
DOI: https://doi.org/10.7554/eLife.41023.015

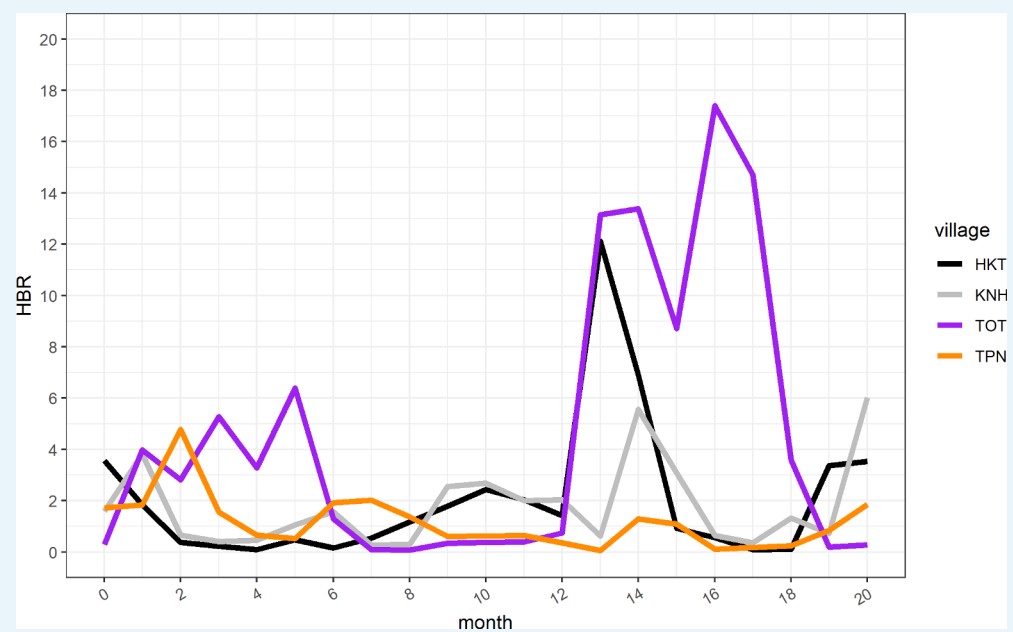

**Appendix 1— figure 3.** Human biting rate (HBR) for primary vectors by study month.
DOI: https://doi.org/10.7554/eLife.41023.016

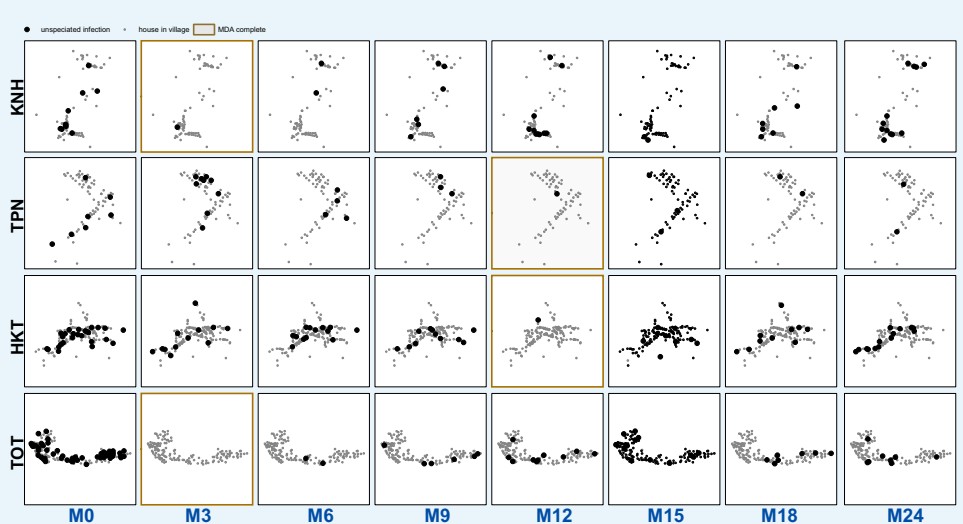

**Appendix 1— figure 4.** Unspeciated uPCR-detected infection for each village and survey.
DOI: https://doi.org/10.7554/eLife.41023.017

## Longitudinal analysis of clinical *P. vivax* episodes

Clinical *P. vivax* episodes exhibited household clustering. Individuals who lived in a house with someone who had a clinical *P. vivax* episode during the same month had nearly six times the odds of having a clinical *P. vivax* episode when compared to those who did not live in a house with someone who had a clinical *P. vivax* episode (AOR: 5.8; CI: 3.4–9.9) (***Appendix 1—table 4***). Clinical *P. vivax* episodes were also most common among the youngest age group (0 to 4 year olds), with those who were age 15 and above having a 60% decrease in the odds of having a clinical episode when compared to the 0 to 4 age group (AOR: 0.4; CI: 0.2–0.9).

**Appendix 1—table 4.** Multivariable mixed effects logistic regression for odds of having a clinical *P. vivax* episode. The model includes a random intercept for individual participants, with repeat observations occurring within individuals over the study period. Study month was included as a control (a linear specification was used, but polynomial specifications were also tested). The covariates for human biting rate (HBR) and having a house member with a clinical episode in the same month were specified as time-varying covariates.

| Covariate | AOR | p-value |
| --- | --- | --- |
| Age 0 to 4 | comparison | |
| Age 5 to 14 | 1.0 (0.4–2.4) | 0.9945 |
| Age 15 plus | 0.4 (0.2–0.9) | 0.0358 |
| female | comparison | |
| male | 0.8 (0.4–1.5) | 0.4687 |
| participated in no rounds of MDA | comparison | |
| participated in MDA (at least one round) | 1.7 (0.7–4.3) | 0.2678 |
| no house member with clinical episode | comparison | |
| house member with clinical episode | 5.8 (3.4–9.9) | <0.0001 |
| low neighborhood non-adherence to MDA | comparison | |
| mid neighborhood non-adherence to MDA | 1.6 (0.3–7.6) | 0.5546 |
| high neighborhood non-adherence to MDA | 1.8 (0.3–9.8) | 0.5205 |
| mean village HBR | 1.0 (1.0–1.1) | 0.1447 |
| village KNH | comparison | |

*Appendix 1—table 4 continued on next page*

*Appendix 1—table 4 continued*

| Covariate | AOR | p-value |
| --- | --- | --- |
| village TPN | 0.6 (0.2–1.8) | 0.3666 |
| village HKT | 0.3 (0.1–1.5) | 0.1329 |
| village TOT | 0.6 (0.1–3.4) | 0.6074 |
| study month | 1.0 (1.0–1.1) | 0.0126 |

DOI: https://doi.org/10.7554/eLife.41023.018

Logistic regression for odds of having a uPCR-diagnosed infection after MDA uPCR detected *P. vivax* infections occurred mostly in older children (AOR: 2.6; CI: 1.7–3.9), adults (AOR: 2.1; CI: 1.4–3.2) and males (AOR: 1.8; CI: 1.4–2.2) (*Appendix 1—table 5*). Individuals who lived in a house with someone else with a uPCR-detected *P. vivax* infection had 1.5 times the odds (CI: 1.2–1.9) of also having a uPCR detected infection after MDA (*Appendix 1—table 5*).

**Appendix 1—table 5.** Multivariable logistic regression for the odds of having a uPCR detected *P. vivax* infection after MDA. Individuals in the data were coded as having an infection of either species if they were ever determined by uPCR to have an infection through blood screenings in full village blood surveys after MDA.

| Covariate | AOR | p-value |
| --- | --- | --- |
| 0 to 4 | comparison | |
| five to 14 | 2.6 (1.7–3.9) | <0.0001 |
| 15 plus | 2.1 (1.4–3.2) | 0.0002 |
| female | comparison | |
| male | 1.8 (1.4–2.2) | <0.0001 |
| participated in no rounds of MDA | comparison | |
| participated in MDA (at least one round) | 0.6 (0.4–0.8) | 0.0027 |
| no house member with uPCR infection | comparison | |
| house member with uPCR infection | 1.5 (1.2–1.9) | 0.0022 |
| no house member with clinical episode | comparison | |
| house member with clinical episode | 1.3 (1.0–1.7) | 0.0284 |
| low neighborhood non-adherence to MDA | comparison | |
| mid neighborhood non-adherence to MDA | 0.6 (0.4–1.1) | 0.0994 |
| high neighborhood non-adherence to MDA | 0.6 (0.4–1.1) | 0.0945 |
| KNH | comparison | |
| TPN | 1.0 (0.7–1.5) | 0.8617 |
| HKT | 1.4 (0.8–2.5) | 0.2614 |
| TOT | 5.4 (2.9–9.8) | <0.0001 |
| number of surveys attended | 1.4 (1.3–1.5) | <0.0001 |

DOI: https://doi.org/10.7554/eLife.41023.019

## Discussion

Post-MDA clinical *P. vivax* episodes exhibited spatiotemporal clustering within houses. uPCR-detected *P. vivax* infections also clustered within houses. While there was an immediate reduction in blood-stage *P. vivax* following MDA (evident in *Appendix 1—figure 1*) there was no overall effect of MDA on the risk of subsequent clinical episodes or uPCR-detected infections over the entire surveillance period.

**Appendix 1—table 6.** Household and population counts for study villages Data used in this analysis are available following the Mahidol-Oxford Tropical Medicine Research Unit data access policy. Both the policy and application form are available at: http://www.tropmedres.ac/data-sharing.

| Village | Households | Population |
|---|---|---|
| KNH | 86 | 504 |
| TPN | 75 | 468 |
| HKT | 176 | 1338 |
| TOT | 149 | 919 |
| total | 486 | 3229 |

DOI: https://doi.org/10.7554/eLife.41023.020

Clinical *P. vivax* episodes occurred more commonly among the youngest age group (0 to 4) while uPCR-detected infections occurred more commonly among adults. This pattern was previously described in Thai-Burma (now referred to as Myanmar) border populations over two decades ago (*Kulldorff, 1997*) and is likely the result of some level of acquired immunity to *P. vivax* infections with time. Both adults and children are exposed to similar levels of *P. vivax* transmission, yet in children the infection is more likely to result in clinical symptoms whereas in adults, many of the infections are likely to become or remain asymptomatic.

Clustering of *P. vivax* across houses persisted across time before and after MDA (*Appendix 1—figure 1*). Overlapping clusters of clinical *P. vivax* episodes and uPCR-diagnosed *P. vivax* infections were observed in KNH (M9 through M15) and TOT (M24). There were overlapping clusters of clinical *P. vivax* episodes and clinical *P. falciparum* episodes in village KNH (M5 – M7) and village TOT (M12 – M18).

