## [Decision Letter]

Thank you for submitting your article "Herd protection against *Plasmodium falciparum* infections conferred by mass antimalarial drug administrations" for consideration by *eLife*. Your article has been reviewed by Prabhat Jha as the Senior Editor, a Reviewing Editor, and three reviewers. The following individual involved in review of your submission has agreed to reveal her identity: Gillian Stresman (Reviewer #3).

The reviewers have discussed the reviews with one another and the Reviewing Editor has drafted this decision to help you prepare a revised submission.

Summary:

This paper presents an interesting dataset on the spatial and temporal patterns of malaria before and after a mass drug administration intervention in 4 villages. Understanding the effect of community-level protection that results from mass drug administration campaigns is currently a key gap in planning for malaria elimination activities. The data are impressively detailed.

Essential revisions:

This paper presents a very interesting dataset on the spatial and temporal patterns of malaria before and after a mass drug administration intervention in 4 villages. Understanding the effect of community-level protection that results from mass drug administration campaigns is currently a key gap in planning for malaria elimination activities. The data are impressively detailed.

However, we have concerns around the presentation of ideas, results, and conclusions with the overall narrative lacking cohesion/focus. For example, the aim of the work is to look at the presence of herd immunity as a result of MDA but the majority of the introduction section focuses on justifying the use of MDA instead of the literature around adherence being a critical factor in the success of MDA, the concept of herd immunity, how critical thresholds for coverage do not exist but potential ways of estimating them etc. Also, the subsequent sections jump between outcome of temporal and spatial analysis, between Pf vs. Pv and infection vs. clinical case making it hard to follow. Overall, the aims, results, and discussion do not match the title and conclusions of paper. We suggest that the authors take time to carefully frame the story they are trying to tell. One suggestion would be to include the Pv work as supplemental information. The outcome of the MDA on Pv is essentially a null result with the MDA implementation not being designed to have an impact on transmission (no radical cure). The findings are important but not surprising and therefore could be included as supplemental so it doesn't detract from the main and more interesting outcome of the impact on Pf.

A second major concern is that the work does not appear to be placed within the larger research and programmatic context. A lot of the research from the GMS region come from this institution, but degree of self-citation is concerning with ~50% of references coming from the research group. Furthermore, the authors appear to be drawing conclusions achieved in other papers without referencing instead of focusing on the aims of this specific piece of work. For example, the Discussion section comments on individual-level results, that were not presented in this paper but have been highlighted in other papers, on the same data without referencing (e.g. "no protective effect.[…] at the individual level"; "those who lived in […] regardless of their individual participation in MDA").

Third, the approach taken to estimate HBR, a key factor suggested to contribute to reintroduction of Pf transmission is questionable. Although a pragmatic approach, what level of spatial bias has been introduced by attributing household-level HBR values based on the nearest trapping site? Given the known high-degree of spatial heterogeneity at small scales in mosquito abundance, would this add an additional level of bias when just selecting the nearest sampling point? Would it be more appropriate to aggregate HBR according to neighborhood as has been defined by the adherence variable to ensure spatial congruence? This can be achieved by creating prediction surfaces using model based geostatistical methods with the HBR variable as outcome and then taking the mean predicted HBR using the same window size as for adherence.

That said, there are many strong points about the analysis e.g. the possibility to look at both individual level and area-level covariates, and the measurement of both symptomatic and asymptomatic infections and relationship between the two.

An interesting result was the apparent clustering of resurgent cases in the area with lower adherence to MDA in one village (apparent smaller herd effect). But the evidence for this was rather weak – it was based mainly on one cluster of non-adherence in one village, which also happened to experience a higher number of cases. This seems potentially explained by the higher biting rate of mosquitoes in this part of the village rather than the non-adherence, but it was not clear whether there was multivariate adjustment to check this? Also, the asymptomatic infections did not seem to follow the same pattern. Including this finding in the title and as a key result elsewhere in the paper places too much confidence in this result, and the conclusions should be more cautious. Perhaps generally the limitations of spatial analysis within only 4 villages could be further acknowledged.

We also suggest the authors explore further the effect of individual level adherence vs area-level average adherence, specifically are non-adherent individuals living in high-adherence neighbourhoods protected, compared with non-adherent individuals in low-adherence neighbourhoods? (and vice versa) It is not really clear at the moment whether this is a spatial effect, or whether it's just perhaps that non-adherent people are both more likely to be infected and to live in non-adherent neighbourhoods (subsection “Spatiotemporal patterns in uPCR-detected infections, clinical episodes and MDA adherence” and subsection “Longitudinal analysis of clinical *P. falciparum* and *P. vivax* episodes”). If a critical cutoff value for adherence levels to achieve a 'herd immunity' during MDA campaigns could be estimated using these data, it would be a major finding and significant improvement to this work. This would be possible to do with these data using ROC analysis with the continuous adherence variables.

Last, in general it was not clear whether multivariate analysis was done – if so, it would be important to see the effect of each variable after adjustment. If not, the authors need to explain why multivariate analysis was not performed.

[Editors' note: further revisions were requested prior to acceptance, as described below.]

Thank you for resubmitting your work entitled "Herd protection against *Plasmodium falciparum* infections conferred by mass antimalarial drug administrations" for further consideration at *eLife*. Your revised article has been favorably evaluated by Neil Ferguson (Senior Editor), a Reviewing Editor, and three reviewers.

The manuscript has been improved but there are some remaining issues that need to be addressed before acceptance, as outlined below:

*Reviewer #1:*

The rewritten paper is much clearer and easier to read than the initial version. Most of the points raised in the previous reviews have been well addressed.

*Reviewer #2:*

The authors have made substantial revisions based on the previous reviews and most points are thoroughly addressed. I like the new Figure 4.

The idea of a herd effect is interesting and certainly worth reporting, but my only remaining concern is that the conclusions and title are too strong based on the limited data from one village and also analytical problems. The authors have clarified in response to another reviewer that their multivariate model (Table 2) does not include random neighbourhood effects, which is not an appropriate analysis with which to test neighbourhood variables since it will overestimate the statistical significance of the herd effect result.

Ideally an alternative analysis needs to be tried, for example separating the village into discrete spatial units and adding random effects at this level. At the very least the authors should consider wording the Abstract and Title more cautiously – to clarify that this herd effect was only seen in one village and more investigation is needed to verify whether this is a real phenomenon.

*Reviewer #3:*

This paper is an interesting and very timely assessment of the community effect of a mass drug administration trial. The revised version of the paper is much improved but could be further strengthened by focusing the Introduction and Discussion section around the objective of the paper, instead of the more general role of MDA as is currently written. Specifically, I suggest that the authors should address the following comments.

Introduction: Are the authors trying to say that MDA should supplement existing malaria programs/used when the full package doesn't eliminate transmission (as is already stated)? There is an argument that this intervention could bypass weak health systems to interrupt transmission and it has been used under alternative scenarios (e.g. Ebola epidemic response). Should this be 'should ideally be used in settings…'

The potential risks associated of using MDA are missing from the introduction (i.e. developing drug resistance). Why has the WHO avoided recommending it for so long? What's changed their stance?

Introduction: The justification that MDA is used to achieve a population level effect is missing important nuances (and offers some repetition). The rationale of MDA is typically argued from the biological perspective whereby the transmission potential/reproductive rate of malaria is so high, to achieve interruption, enough of the parasitic reservoir needs to be removed before it has the chance to fuel subsequent infections. The concept of a "herd effect" applies in most communicable disease systems: If enough of the population is protected, transmission chains cannot be sustained and transmission with die out. What is 'enough' is unknown as stated and is affected by the propensity for diseases to cluster in populations (defined either spatially or demographically). If the aim of MDA for malaria is to interrupt transmission chains, the notion of a herd effect providing additional levels of population protection is plausible but has never been examined empirically.

Subsection “Analysis”: what was the rationale for using a 100m radius. Are the results affected according to size of buffer?

Subsection “Analysis”: is there an impact when considering individual level (e.g. proportion of individuals non-compliant) vs household level (e.g. households with </> 80%? compliance) non-compliance?

Discussion section: MDA adherents in low-adherence areas had highest risk? Even compared to those in the same neighborhoods with low-adherence? This is an interesting finding consistent with what has been observed with bednets: those who have the most malaria are more likely to adhere to interventions, potentially related to perceptions of risk.

Discussion section: I would encourage authors to split this paragraph into 3 sections to ensure that these key observations can be discussed in depth.

Discussion section: do the authors have evidence to support this counterfactual? Or do they mean would have likely/may have resulted in… ?

Discussion section: I'm not sure of the relevant of this paragraph to the objectives of the work. The focus is on the herd protection or am I missing a connection between adherence and the more general rational for MDA? What are the potential limitations to MDA when considering the potential impacts of herd protection that may be specific to this setting?

Discussion section: Similar to point above, the objective is the work is related to herd immunity and the concluding remarks seem to focus on the general impact of MDA. Here, I would encourage the authors to include some insight revolving around the potential that comes with an improved understanding of the herd effect for malaria control and elimination activities. Next steps for further exploring this phenomenon?

Discussion section: The objective of the study was to look at the herd effect and the impact of non-adherence. I would encourage the authors to re-write this first paragraph of the discussion to focus on this instead of what has been reported in other publications.

[Editors' note: further revisions were requested prior to acceptance, as described below.]

Thank you for resubmitting your work entitled "Potential herd protection against *Plasmodium falciparum* infections conferred by mass antimalarial drug administrations" for further consideration at *eLife*. Your revised article has been favorably evaluated by Neil Ferguson (Senior Editor), a Reviewing Editor, and 3 reviewers.

The manuscript has been improved but there are some minor remaining issues that need to be addressed before acceptance, as outlined below in the individual reports.

*Reviewer #1:*

I had no substantive concerns with the previous version and feel that the manuscript is essentially suitable for publication.

However, in the course of the review process the authors have made incremental amendments that understate the very real concerns that exist around the potential for selection of resistance by MDA. In particular, reference 18 (one of the authors' own publications, which is not so easy to access because of a paywall) is cited to support the contention that there is "no evidence of resistance emerging as a result of controlled, targeted MDA with the use of therapeutic drug levels". Arguable, such evidence is exactly what was reported by Trigg et al. for pyrimethamine/sulfadoxine in Tanzania (Trig.et al., 1997). Even if there were no evidence, the absence of evidence is not good evidence for absence of an effect, and the case for no effect seems to be summarised in the Abstract of von Seidlein and Dondorp 2015 by the statement "it is difficult to conceptualize how targeted malaria elimination could contribute to artemisinin resistance…". The wording here is critical: it is rather easy to conceptualise how failed attempts at elimination using MDA could lead to resistance, especially if drug use is not well controlled. Since this paper is not about drug resistance, I think that the claims about resistance in the Introduction should be removed.

*Reviewer #2:*

I copy here my previous review and the authors' response, with my current response below:

“The authors have made substantial revisions based on the previous reviews and most points are thoroughly addressed. I like the new Figure 4.

The idea of a herd effect is interesting and certainly worth reporting, but my only remaining concern is that the conclusions and title are too strong based on the limited data from one village and also analytical problems. The authors have clarified in response to another reviewer that their multivariate model (Table 2) does not include random neighbourhood effects, which is not an appropriate analysis with which to test neighbourhood variables since it will overestimate the statistical significance of the herd effect result.

Ideally an alternative analysis needs to be tried, for example separating the village into discrete spatial units and adding random effects at this level. At the very least the authors should consider wording the abstract and title more cautiously – to clarify that this herd effect was only seen in one village and more investigation is needed to verify whether this is a real phenomenon.”

“Yes, this is an approach used, for example, by Ali et al. (2005, Lancet, "Herd immunity conferred by killed oral cholera vaccines in Bangladesh") and in some ways it would be ideal. With our data some problems would remain though because any discrete spatial units ("neighborhoods") that are imposed on the villages would be arbitrary and since the villages are quite small, there would only be 4-5 each (unless using very small "neighborhoods"). For the one village with Pf after MDA, we'd only have 4 or 5 neighborhoods and we would question if a random intercept would actually be useful, given that there are assumptions about the mean and variance of the random intercept.

There is also a spatial argument to be made in that an arbitrary boundary or line that would split such areas within the village would be creating a boundary between houses that are extremely close to each other. If geographic proximity is important here, and we believe it is, then we would actually lose a lot of information by using this approach.

Regardless, the reviewer is correct that we need to be cautious in our wording here. We have taken the suggestion to word the Title and Abstract more cautiously, and we discuss some of these issues in the Discussion section.”

Thank you for addressing these points very well in terms of altering the abstract and title. For my last suggestion, please can the authors either (a) explicitly state that the confidence interval reported for the paper's main result is likely to be much too narrow, because it tests for the effect of a neighbourhood level variable (coverage within 100m of a household) using individual level data without allowing for neighbourhood effects. Or (b) remove the confidence interval entirely and explain why it could not be calculated. Or (c) try an alternative analysis.

I appreciate the difficulties of doing an analysis which incorporates neighbourhood effects, but the confidence interval of the main result in the Abstract, shown in subsection “Longitudinal multivariable analysis of clinical *P. falciparum* episodes” as it stands must therefore be incorrect (AOR: 2.85; CI: 1.28 – 6.37). It would be unlikely to be a significant result if neighbourhood effects were included. It is still an interesting result, which could be explored further in other larger datasets.

Otherwise the manuscript is very interesting and useful, and I have no further comments.

Reviewer #3:

This work is much improved. I have no additional comments and recommend the work for publication.

---

## [Author Response]

Essential revisions:[…] 1) However, we have concerns around the presentation of ideas, results, and conclusions with the overall narrative lacking cohesion/focus. For example, the aim of the work is to look at the presence of herd immunity as a result of MDA but the majority of the introduction section focuses on justifying the use of MDA instead of the literature around adherence being a critical factor in the success of MDA, the concept of herd immunity, how critical thresholds for coverage do not exist but potential ways of estimating them etc. Also, the subsequent sections jump between outcome of temporal and spatial analysis, between Pf vs. Pv and infection vs. clinical case making it hard to follow. Overall, the aims, results, and discussion do not match the title and conclusions of paper. We suggest that the authors take time to carefully frame the story they are trying to tell. One suggestion would be to include the Pv work as supplemental information. The outcome of the MDA on Pv is essentially a null result with the MDA implementation not being designed to have an impact on transmission (no radical cure). The findings are important but not surprising and therefore could be included as supplemental so it doesn't detract from the main and more interesting outcome of the impact on Pf.

We have now refocused and rewritten the manuscript in order to address this valid critique. The main goal of this work was to analyze clinical *P. falciparum* malaria after MDA had been administered, looking at both individual and group-level effects. Since the work was focused on clinical P. falciparum we’ve now removed all analysis of *P. vivax* and uPCR-detected *P. falciparum* and offer these analyses in the supplementary material. We’ve rewritten the Introduction to focus more generally on MDA for *P. falciparum* malaria and have revised much of the rest of the manuscript in order to improve readability. We believe that these changes make the manuscript much more clear and concise.

2) A second major concern is that the work does not appear to be placed within the larger research and programmatic context. A lot of the research from the GMS region come from this institution, but degree of self-citation is concerning with ~50% of references coming from the research group. Furthermore, the authors appear to be drawing conclusions achieved in other papers without referencing instead of focusing on the aims of this specific piece of work. For example, the Discussion section comments on individual-level results, that were not presented in this paper but have been highlighted in other papers, on the same data without referencing (e.g. "no protective effect […] at the individual level"; "those who lived in [.…] regardless of their individual participation in MDA").

In our refocusing of this work, we now begin by discussing MDA in general rather than the GMS context. We’ve now attempted to appropriately reference other research teams in this revision.

Regarding results from other manuscripts – these data have been previously analyzed (Landier et al., 2017). However, the effects that we mention (for example individual versus group effects from MDA) are from this analysis. The analysis of risk factors for acquiring a clinical *P. falciparum* episode after MDA (results in Table 2) uses a multivariable regression which includes both individual and group (“neighborhood”) level variables for MDA adherence. We hope that this is now made clear through our revision of the manuscript.

3) Third, the approach taken to estimate HBR, a key factor suggested to contribute to reintroduction of Pf transmission is questionable. Although a pragmatic approach, what level of spatial bias has been introduced by attributing household-level HBR values based on the nearest trapping site? Given the known high-degree of spatial heterogeneity at small scales in mosquito abundance, would this add an additional level of bias when just selecting the nearest sampling point? Would it be more appropriate to aggregate HBR according to neighborhood as has been defined by the adherence variable to ensure spatial congruence? This can be achieved by creating prediction surfaces using model based geostatistical methods with the HBR variable as outcome and then taking the mean predicted HBR using the same window size as for adherence.

The reviewer is correct, both about micro-heterogeneities in HBR and that our specification of this variable is overly simplistic. We agree that a smoothed surface would be better, however we don’t believe that these data are appropriate for making prediction surfaces. There were only 5 catch sites per village and most villages are not in geographic proximity to each other (therefore not appropriate to make a single surface for all villages). The catch sites are not systematically distributed through the villages and their geographic extent is not as broad as the geographic extent of houses in each village. We would therefore have to create interpolations from only 5 points and then extrapolate outside of the 5 points to meet the geographic range of the villages. To address this problem in our analysis we have therefore opted to use HBR calculations at the village level in the regressions. This is still simplistic and doesn’t address micro-scale heterogeneities. In the absence of detailed human movement data and HBR outside of the village, we believe that this specification for HBR is the best possible approach with these data.

4) An interesting result was the apparent clustering of resurgent cases in the area with lower adherence to MDA in one village (apparent smaller herd effect). But the evidence for this was rather weak – it was based mainly on one cluster of non-adherence in one village, which also happened to experience a higher number of cases. This seems potentially explained by the higher biting rate of mosquitoes in this part of the village rather than the non-adherence, but it was not clear whether there was multivariate adjustment to check this? Also, the asymptomatic infections did not seem to follow the same pattern. Including this finding in the title and as a key result elsewhere in the paper places too much confidence in this result, and the conclusions should be more cautious. Perhaps generally the limitations of spatial analysis within only 4 villages could be further acknowledged.

Please see our response to point 2 above. We hope that the revised manuscript now clarifies that the models were multivariable and are looking at both individual and group level MDA adherence as well as HBR.

The reviewer is correct that these are a limited set of data and that we should be cautious in over-interpreting the results (which mostly come from a single village that had remaining Pf after MDA). We have added the following statement to the Discussion section:

“Finally, these data come from a limited number of villages (total of 4), with analysis of *P. falciparum* episodes coming from the sole village that continued to have *P. falciparum* after MDA.”

5) We also suggest the authors explore further the effect of individual level adherence vs area-level average adherence, specifically are non-adherent individuals living in high-adherence neighbourhoods protected, compared with non-adherent individuals in low-adherence neighbourhoods? (and vice versa) It is not really clear at the moment whether this is a spatial effect, or whether it's just perhaps that non-adherent people are both more likely to be infected and to live in non-adherent neighbourhoods (subsection “Spatiotemporal patterns in uPCR-detected infections, clinical episodes and MDA adherence” and subsection “Longitudinal analysis of clinical P. falciparum and P. vivax episodes”).

Please see our response to point 2 above. We hope that the revised manuscript now clarifies that the models were multivariable and are looking at both individual and group /area level adherence and effects.

6) If a critical cutoff value for adherence levels to achieve a 'herd immunity' during MDA campaigns could be estimated using these data, it would be a major finding and significant improvement to this work. This would be possible to do with these data using ROC analysis with the continuous adherence variables.

Our goal in this paper is to show that there is a measurable herd effect, but we believe that it is drastically complicated by context and that a single proportion of adherence won’t be applicable in all target populations. This analysis will soon be followed by a mathematical modelling paper using these and other data in order to further explore this topic.

7) Last, in general it was not clear whether multivariate analysis was done – if so, it would be important to see the effect of each variable after adjustment. If not, the authors need to explain why multivariate analysis was not performed.

Please see our response to point 2 above. We hope that the revised manuscript now clarifies that the models were multivariable and are looking at both individual and group /area level adherence and effects.

[Editors' note: further revisions were requested prior to acceptance, as described below.]

Reviewer #2:[…] The idea of a herd effect is interesting and certainly worth reporting, but my only remaining concern is that the conclusions and title are too strong based on the limited data from one village and also analytical problems. The authors have clarified in response to another reviewer that their multivariate model (Table 2) does not include random neighbourhood effects, which is not an appropriate analysis with which to test neighbourhood variables since it will overestimate the statistical significance of the herd effect result.Ideally an alternative analysis needs to be tried, for example separating the village into discrete spatial units and adding random effects at this level. At the very least the authors should consider wording the Abstract and Title more cautiously – to clarify that this herd effect was only seen in one village and more investigation is needed to verify whether this is a real phenomenon.

Yes, this is an approach used, for example, by Ali et al., (2005) and in some ways it would be ideal. With our data some problems would remain though because any discrete spatial units (“neighborhoods”) that are imposed on the villages would be arbitrary and since the villages are quite small, there would only be 4-5 each (unless using very small “neighborhoods”). For the one village with Pf after MDA, we’d only have 4 or 5 neighborhoods and we would question if a random intercept would actually be useful, given that there are assumptions about the mean and variance of the random intercept.

There is also a spatial argument to be made in that an arbitrary boundary or line that would split such areas within the village would be creating a boundary between houses that are extremely close to each other. If geographic proximity is important here, and we believe it is, then we would actually lose a lot of information by using this approach.

Regardless, the reviewer is correct that we need to be cautious in our wording here. We have taken the suggestion to word the title and abstract more cautiously, and we discuss some of these issues in the Discussion section.

Reviewer #3:[…] Introduction: Are the authors trying to say that MDA should supplement existing malaria programs/used when the full package doesn't eliminate transmission (as is already stated)? There is an argument that this intervention could bypass weak health systems to interrupt transmission and it has been used under alternative scenarios (e.g. Ebola epidemic response). Should this be 'should ideally be used in settings…'

Yes. We have now changed the wording of this statement (Introduction).

The potential risks associated of using MDA are missing from the introduction (i.e. developing drug resistance). Why has the WHO avoided recommending it for so long? What's changed their stance?

We have now added a few sentences discussing historical problems and concerns regarding MDA:

Introduction

“Given that drug pressure (through provision of antimalarial drugs) provides a survival advantage for resistant parasites, there has been some hesitance in using MDA for malaria. One historical malaria eradication campaign relied on the inclusion of sub-therapeutic levels of antimalarials distributed in table salt across large populations [15]. This program likely led to the emergence of parasite resistance in the same regions [16] and this has in part led to hesitance among some institutions (i.e. the World Health Organization and ministries of health) to implement MDA for malaria [17]. There is, however, no evidence of resistance emerging as a result of controlled, targeted MDA with the use of therapeutic drug levels [18].”

Introduction: The justification that MDA is used to achieve a population level effect is missing important nuances (and offers some repetition). The rationale of MDA is typically argued from the biological perspective whereby the transmission potential/reproductive rate of malaria is so high, to achieve interruption, enough of the parasitic reservoir needs to be removed before it has the chance to fuel subsequent infections. The concept of a "herd effect" applies in most communicable disease systems: If enough of the population is protected, transmission chains cannot be sustained and transmission with die out. What is 'enough' is unknown as stated and is affected by the propensity for diseases to cluster in populations (defined either spatially or demographically). If the aim of MDA for malaria is to interrupt transmission chains, the notion of a herd effect providing additional levels of population protection is plausible but has never been examined empirically.

We have now attempted to incorporate these nuances and some of this text (reworded) into this part of the manuscript:

Introduction

“While antimalarials are usually administered following diagnosis (confirmed or presumed) or used as a prophylactic, MDA is used because of an intended population- or community-level effect. […] If the aim for of antimalarial MDA is to interrupt transmission, the notion of a herd effect providing additional levels of population protection is plausible but has not been examined empirically [22].”

Subsection “Analysis”: what was the rationale for using a 100m radius. Are the results affected according to size of buffer?

The 100m radius was relatively arbitrary, but we did do a sensitivity analysis on the buffer size. We’ve added this detail in the Appendix now:

“The sensitivity of the model results with regard to the neighborhood MDA non-adherence buffer size was also tested. Neighborhood MDA-non-adherence was tested with varying buffer sizes, beginning with a radius of 20m and going up to a radius of 220m, by 40m intervals. Neighborhood buffers from 60m to 140m were statistically significant and had comparable model adjusted odds ratios (at 60m: AOR = 2.33; CI: 1.03 – 5.25; at 100m: AOR = 2.84; CI: 1.29 – 6.27; at 140m: AOR = 2.35; CI: 1.06 – 5.19). Small buffers (i.e. 20m) included only one or very few households. At larger buffer sizes (> 180m) most houses were included within a buffer (furthest distance between any two houses was approximately 1km) leading to little spatial heterogeneity in participation across the village.”

Subsection “Analysis”: is there an impact when considering individual level (e.g. proportion of individuals non-compliant) vs household level (e.g. households with </> 80%? compliance) non-compliance?

For this analysis we’ve only considered the individual level. We prefer to not do the analysis using the household level because of the variation in household size (histogram attached here). For example, a house with 10 members and 80% adherence could still have 2 members with parasites in their blood while a house with 5 members and only 60% adherence would still have 2 members with parasites in their blood. Arguably, these 2 individuals are equally infective for others in the neighborhood, but would be treated as though they were completely different if we analyzed based on the house level.

Discussion section: MDA adherents in low-adherence areas had highest risk? Even compared to those in the same neighborhoods with low-adherence? This is an interesting finding consistent with what has been observed with bednets: those who have the most malaria are more likely to adhere to interventions, potentially related to perceptions of risk.

We have now added the following statement along these lines:

Discussion section

“As has been described in other settings, this individual-level finding may be related to relative perceptions of risk; with potential complacency among individuals living in areas with lower levels of malaria [23].”

Discussion section: I would encourage authors to split this paragraph into 3 sections to ensure that these key observations can be discussed in depth.

Done.

Discussion section: do the authors have evidence to support this counterfactual? Or do they mean would have likely/may have resulted in….

We have softened the statement as requested (Discussion section).

Discussion section: I'm not sure of the relevant of this paragraph to the objectives of the work. The focus is on the herd protection or am I missing a connection between adherence and the more general rational for MDA? What are the potential limitations to MDA when considering the potential impacts of herd protection that may be specific to this setting?

We have now dropped this paragraph.

Discussion section: Similar to point above, the objective is the work is related to herd immunity and the concluding remarks seem to focus on the general impact of MDA. Here, I would encourage the authors to include some insight revolving around the potential that comes with an improved understanding of the herd effect for malaria control and elimination activities. Next steps for further exploring this phenomenon?

We have now rewritten the conclusion to focus more specifically on herd effects from MDA for *P. falciparum*, general implications of this work, and future directions.

Discussion section: The objective of the study was to look at the herd effect and the impact of non-adherence. I would encourage the authors to re-write this first paragraph of the discussion to focus on this instead of what has been reported in other publications.

Done.

[Editors' note: further revisions were requested prior to acceptance, as described below.]

Reviewer #1:[…] However, in the course of the review process the authors have made incremental amendments that understate the very real concerns that exist around the potential for selection of resistance by MDA. In particular, reference 18 (one of the authors' own publications, which is not so easy to access because of a paywall) is cited to support the contention that there is "no evidence of resistance emerging as a result of controlled, targeted MDA with the use of therapeutic drug levels". Arguable, such evidence is exactly what was reported by Trigg et al. for pyrimethamine/sulfadoxine in Tanzania (Trig et al., 1997). Even if there were no evidence, the absence of evidence is not good evidence for absence of an effect, and the case for no effect seems to be summarised in the Abstract of von Seidlein and Dondorp 2015 by the statement "it is difficult to conceptualize how targeted malaria elimination could contribute to artemisinin resistance…". The wording here is critical: it is rather easy to conceptualise how failed attempts at elimination using MDA could lead to resistance, especially if drug use is not well controlled. Since this paper is not about drug resistance, I think that the claims about resistance in the Introduction should be removed.

We were unsure of whether or not the reviewer wanted us to discard all discussion/claims about resistance in the paragraph – but we agree with a previous suggestion that the hesitancy for using MDA should at least be mentioned. We felt that the most contentious statement was the one referred to above (“with no evidence of resistance emerging as a result of controlled, targeted MDA…”) and we have therefore removed this statement from the manuscript.

Reviewer #2:[…] For my last suggestion, please can the authors either (a) explicitly state that the confidence interval reported for the paper's main result is likely to be much too narrow, because it tests for the effect of a neighbourhood level variable (coverage within 100m of a household) using individual level data without allowing for neighbourhood effects. Or (b) remove the confidence interval entirely and explain why it could not be calculated. Or (c) try an alternative analysis.I appreciate the difficulties of doing an analysis which incorporates neighbourhood effects, but the confidence interval of the main result in the Abstract, shown in subsection “Longitudinal multivariable analysis of clinical P. falciparum episodes” as it stands must therefore be incorrect (AOR: 2.85; CI: 1.28 – 6.37). It would be unlikely to be a significant result if neighbourhood effects were included. It is still an interesting result, which could be explored further in other larger datasets.

Done. We have included the following statement at the end of the Discussion section, immediately before stating our conclusions:

“Given that clinical *P. falciparum* episodes post-MDA were only possible to analyze in a single village, and that neighborhoods were not discrete and overlapped (100m buffer around each house), a neighborhood-level effect was not included in this analysis. It is possible that the confidence intervals around the neighborhood MDA adherence variable are therefore too small and would not have been statistically significant had a neighborhood effect been included. This work would benefit from analyses with larger datasets.”